# Deep neural networks to recover unknown physical parameters from oscillating time series

**Antoine Garcon** [1,2]*, **Julian Vexler**[1], **Dmitry Budker**[1,2,3], **Stefan Kramer**[1]

**1** Johannes Gutenberg-Universität, Mainz, Germany, **2** Helmholtz-Institut, GSI Helmholtzzentrum für Schwerionenforschung GmbH, Mainz, Germany, **3** Department of Physics, University of California Berkeley, Berkeley, California, United States of America

* garcon@uni-mainz.de

**Data Availability Statement:** All code needed to evaluate the conclusions of this article are present in the article and/or the Supplementary Materials.

**Funding:** This work was supported in part by the Cluster of Excellence PRISMA+ funded by the German Research Foundation (DFG) within the

## Abstract

Deep neural networks are widely used in pattern-recognition tasks for which a human-comprehensible, quantitative description of the data-generating process, cannot be obtained. While doing so, neural networks often produce an abstract (entangled and non-interpretable) representation of the data-generating process. This may be one of the reasons why neural networks are not yet used extensively in physics-experiment signal processing: physicists generally require their analyses to yield quantitative information about the system they study. In this article we use a deep neural network to disentangle components of oscillating time series. To this aim, we design and train the neural network on synthetic oscillating time series to perform two tasks: a *regression* of the signal latent parameters and *signal denoising* by an *Autoencoder*-like architecture. We show that the regression and denoising performance is similar to those of least-square curve fittings with true latent-parameters initial guesses, in spite of the neural network needing no initial guesses at all. We then explore various applications in which we believe our architecture could prove useful for time-series processing, when prior knowledge is incomplete. As an example, we employ the neural network as a preprocessing tool to inform the least-square fits when initial guesses are unknown. Moreover, we show that the regression can be performed on some latent parameters, while ignoring the existence of others. Because the *Autoencoder* needs no prior information about the physical model, the remaining unknown latent parameters can still be captured, thus making use of partial prior knowledge, while leaving space for data exploration and discoveries.

## 1 Introduction

Deep neural networks (DNNs) have been successfully used in a wide variety of tasks, such as regression, classification (e.g, in image or speech recognition [1, 2]), and time-series analysis. They are known for being able to construct useful higher-level features from lower-level

German Excellence Strategy (Project ID 39083149), by the European Research Council (ERC) under the European Union Horizon 2020 research and innovation program (project Dark-OST, grant agreement No 695405), and by the DFG Reinhart Koselleck project. A.G. acknowledges funding from the Emergent AI Center funded by the Carl-Zeiss-Stiftung. The funders had no role in study design, data collection and analysis, decision to publish, or preparation of the manuscript.

**Competing interests:** The authors have declared that no competing interests exist.

features in many applications. However, these feature representations frequently remain incomprehensible to humans.

This property is one of the reasons why DNNs are not more widely used in physics, in which the approach to data exploration is usually drastically different.

Most systems studied in physics are well described by physicals models, generally referred to as *equations of motion*. The experimental data are analysed with respect to a particular model. When doing so, the equations of motion are analytically or numerically solved, yielding a theoretical description of the data-generating process. The resulting model generally includes a set of mathematical variables that can be adjusted to span the data. The true values of these variables are generally unknown and must be recovered. For that reason, we refer to them as *latent parameters*. The true latent parameters are approximated by comparing the data to the model, typically by fitting the model to the data. With this in mind, the ability of DNNs to find abstract representations of the data features rather than a quantitative generating process is generally seen as a limitation rather than an advantage by physicists. For that reason, DNNs are still often viewed as black boxes in physics and started to be used in the field only in recent years [3].

We find this to be a missed opportunity for the physics community. With physical models at hand, one can generate arbitrarily large volumes of synthetic data to train the DNNs, and later process real-world signals [4]. This circumvents many challenges of supervised learning during which DNNs are trained with data for which the true latent parameters (labeled data) need to be known.

Making full use of this possibility, DNNs were recently trained on synthetic nuclear magnetic resonance (NMR) spectroscopic data, simulated by accurate physical models [5]. The large amount of labeled data generated this way enables convergence of the DNN, which is then used to process real NMR data with great accuracy. A similar approach, that is, starting training with synthetic data and continuing with real-world data, has become popular in robotics and autonomous driving.

Moreover, extensive work was done in order to disentangle and make sense of DNN representations. A notable example is that of the $\beta$-variational autoencoder architecture [6]. Correlation loss penalties can also be used during DNN training, without prior knowledge of the data-generating process [7, 8]. These methods consist of penalizing the DNN if its feature representation becomes entangled during training. While doing so, the DNN is encouraged to produce an efficient or disentangled feature representation. While disentangled, the representations achieved through these methods are not readily interpretable and usually require further analysis.

Nonetheless, DNNs are being increasingly used in physics data processing, in particular for signal classification—during which unusual datasets are flagged for further analysis. It was shown that *Autoencoders* can effectively be trained on Large Hadron Collider particle-jet data to detect events or anomalies [9]. In this instance, the DNN is successfully able to increase the events' signal-to-noise ratio by a factor 6. Other searches in high-energy physics, including [10, 11], have recently been performed also with the aim of detecting data displacement from a null-hypothesis (no anomalies). All these searches seek to perform data analyses in a model-independent setting, that is, with minimal prior information or bias. More recently, DNNs have been applied to time-series processing in nano-NMR [12]. In nano-NMR settings, the noise model is complex and noise overpowers the weak signals, rendering standard data analyses inefficient. The DNN was tasked to classify signals (i.e. discriminating two frequencies) and outperformed full-Bayesian methods.

While often achieving great successes, to our knowledge most applications of DNNs in physics are geared toward classification problems. In addition, DNNs are still rarely employed

for time-series analyses, although they are the most common form of data acquired during physics experiments. In this article, we propose to use a DNN to disentangle components of monochromatic, amplitude- and frequency-modulated sine waves (AM/FM-sine waves respectively), arguably the most prevalent forms of time-domain signals in physics. The method yields similar performance as more standard analyses such as least-square curve fittings (LS-fits), during which the data-generating process is assumed to be known and a least-squares regression is performed to predict the signal's latent parameters.

LS-fits, however, require the user to input latent-parameters initial guesses prior to regression. These initial guesses are the prior estimation of the true latent parameters and provide a starting point for the LS-fit gradient descent. The trained DNN, however, needs no initial guesses, thus requiring less prior information about the data-generating process. Indeed, we show that, precisely because DNNs find abstract data representations, they can be used in settings when prior knowledge exists, but is not complete, as it is particularly the case in "new-physics" searches [13], thus leaving space for data exploration and discoveries.

The first part of this article describes the synthetic data that we generate and use throughout this work, i.e. monochromatic, AM- and FM-sine waves time series, and their relevance to real-world physics experiments. We then describe our DNN architecture, which incorporates two tasks: A *Regressor* DNN performs a regression of the signal's latent parameters that are known to be present in the data-generating process. We note here that throughout the paper the term *Regressor* is employed as opposed to a *Classifier*, not as an independent variable in a regression. Therefore, *Regressor* refers to the DNN predicting the signal's latent parameters. In addition, an *Autoencoder* [14] denoises the signals by learning an approximation of the unknown latent parameters. As a benchmarking method, we evaluate the DNN by comparing its performance to an LS-fit with true initial guesses.

We later employ the DNN in realistic settings, when prior knowledge about the data-generating process is incomplete: LS-fit fidelity is typically highly sensitive to initial guesses, thus requiring the user to perform preprocessing work or to possess prior information in order to perform optimally. As a first application, we show that the DNN can be used to predict initial guesses for the model fit evaluation. While consistently converging to optimal solutions, the technique circumvents the usual difficulties arising from fitting signals, such as the need for initial-guesses exploration.

Next, we show that the DNN can be used when the user ignores if the time-series are monochromatic-, AM- or FM-sine waves, but still wishes to recover their main frequency component. In such settings, the user is generally required to repeat the analysis by exploring the space of data-generating processes and initial guesses. Using our architecture enables the user to input only the known information when performing the analysis. That is, the *Regressor* is tasked to recover the user-expected latent parameters while ignoring the existence of others. Because the *Autoencoder* needs no prior information, it is still able to capture unknown information.

## 2 Results

### 2.1 Data description

The time series studied throughout the article are exponentially decaying monochromatic, FM- and AM-sine waves. Gaussian noise is linearly added to the pure signals. An example of FM-signal is shown in Fig 1 (top) alongside its sub-components (decaying-sinewave carrier, frequency-modulation signal, and noise).

Decaying monochromatic-sine waves appear and are prevalent in all fields of physics. They arise from solving the equations of motion of the two-level quantum system, or of the classical

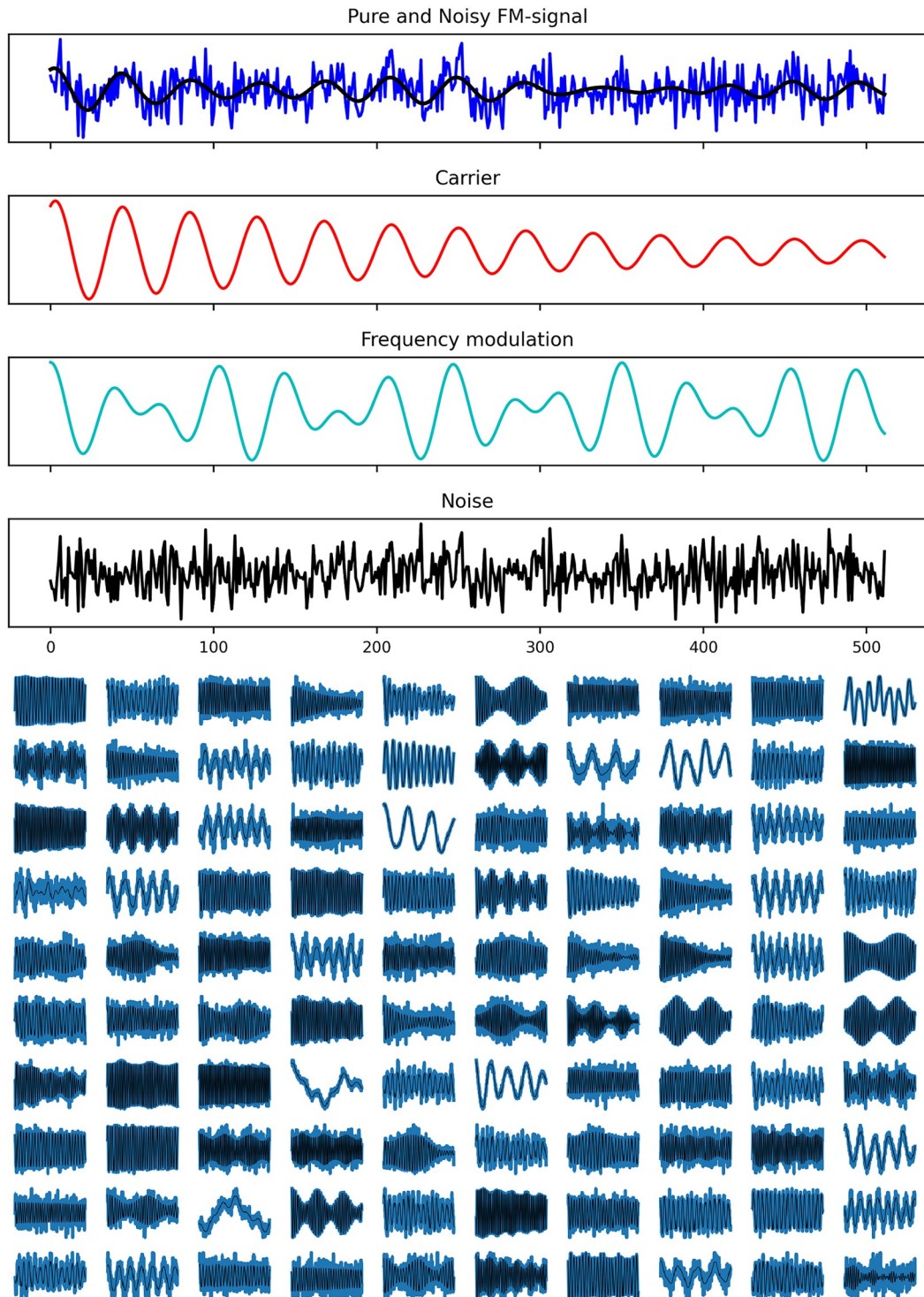

**Fig 1. Examples of frequency-modulated sine wave (FM) synthetic time series. Top**: pure and noisy FM-sine wave decomposition. Gaussian noise and frequency modulation are linearly added to decaying sine wave carrier. **Bottom**: random selection of noisy-input (blue) and pure-target (black) samples, illustrating the effect of the random latent parameter selection.

harmonic oscillator; to which a multitude of other physical systems can be mathematically reduced to. Notorious examples include the spin-1/2 particle in a DC magnetic field, the orbital motion of planets, or RLC circuits. In information theory, the two-level quantum system also provides a complete description of the qbit. Frequency and amplitude modulation generally arise from external factors such as oscillating magnetic or electric fields applied by the experimenters. Amplitude and frequency modulation of a carrier frequency are also the most common scheme of information communication links. Some form of Gaussian noise, while not necessarily always dominant, is in general present in any real-world signal. The statistical Gaussian noise formalism provides an accurate description of electronic thermal-noise, quantum shot noise, black-body radiation, and of White noise in general.

All time series used throughout the article are 512 s long, sampled once per second. The latent parameters used to generate the monochromatic sinewaves are the carrier frequency, $F_c$ and phase $\phi$, in addition to the coherence time $\tau$. The AM- and FM-sinewaves are generated by adding a modulation function to the carrier. The modulation function's latent parameters are the modulation frequency and amplitude, $F_m$ and $I_m$, respectively. Noise is linearly added to the pure signals by sampling the Gaussian distribution with zero mean and standard deviation $\sigma$. The mathematical descriptions of the monochromatic, AM- and FM-sine waves in addition to the data-generation procedure are given in the Methods section of this article.

Before each sample generation, the latent parameters are randomly and uniformly sampled within their respective allowed range (see Methods). The range of $F_c$ ensures the carrier frequency remains well within the Fourier and Nqyist limits such that no over- or under-sampling occurs. The modulation amplitude range ensures the majority of the signal's power remains in its first sidebands and carrier.

Despite requiring only 6 latent parameters to generate the samples, these ranges enable a wide scope of functions to be realized. AM/FM-signals with minimum $I_m$ reduce to decaying monochromatic-sine waves and reach 100% modulation with maximum $I_m$. The coherence time range is wide enough to span underdamped signals up to virtually non-decaying signals. These latent parameter ranges are wide enough such that they would encompass many foreseeable real-world signals. A random selection of FM-signals with and without noise is shown in Fig 1 (bottom), illustrating the richness of the data in a more qualitative manner.

The choice of studying monochromatic, AM-, and FM-sine waves is not only motivated by their richness and prevalence in real-world physics experiments. Indeed, despite originating from different physical models and having different mathematical descriptions, the time series share similar visual features. As a result, within some range of parameters, even expert users could mistake the three generating processes. This is especially the case for weak modulations in the presence of noise, for which visual discrimination in time- or frequency-domain (inspecting the spectrum) may be impossible. For all the reasons cited above, monochromatic-, AM- or FM-sine waves appear as good representative signals on which to perform our study. Nevertheless, the methods presented in this article can be applied to other types of signals as well.

## 2.2 Deep neural network architecture

The latent-parameters regression and signal denoising are performed by two separate architectures described in the Methods section of this article, alongside the Python code implementation. Python code implementation is given in the S1 Appendix.

Denoising is performed by an *Autoencoder* architecture [14] composed of an *Encoder* followed by a *Decoder*. Noisy signals are first passed through the *Encoder*. The *Encoder* output layer has 64 neurons and thus produces a compressed representation of the input signal. Following this step, the *Encoder* output is passed through the *Decoder*, which decompresses the

signal to its original size. This type of [*Encoder-Decoder*] architecture, is widely used, inter alia, for data denoising [15]. As the *Encoder* output dimension is smaller than the dimension of the input data, the *Encoder*'s output layer acts as an information bottleneck, or more specifically dimensionality reduction, thus encouraging the *Autoencoder* to capture relevant latent features while discarding noise or redundant information [14]. Latent-parameters regression is also performed while passing the data through the *Encoder*. The *Encoder* output is then passed through a third DNN referred to as the *Regressor*.

After refining the base *Encoder*, *Regressor* and *Decoder*, we unify the three architectures into a single DNN such that the *Regressor* and *Decoder* share the same *Encoder*. We find that unification is best achieved by merging them into a single DNN as depicted in Fig 2.

The latent-parameter regression loss, $MSE_{reg}$, and signal-denoising losses, $MSE_{dec}$, are computed simultaneously. $MSE_{reg}$ is the mean squared error between the DNN predictions and true latent parameters, while $MSE_{dec}$ is the mean squared error between the denoised signal prediction and the noiseless signal. Finally, the total loss used during backpropagation is computed as a weighted sum of $MSE_{reg}$ and $MSE_{dec}$ as follows:

$$\mathcal{L}oss = \beta \cdot MSE_{reg} + (1 - \beta) \cdot MSE_{dec}, \ \beta \in [0, 1], \tag{1}$$

where the hyperparameter $\beta$ is the bias adjustment between the two tasks.

The value of $\beta$ is determined by performing a grid search over the [0, 1] range: We train a new instance of the DNN for varying values of $\beta$. The best value (yielding the lowest overall loss and low bias towards any of the tasks, determined on a validation set) is maintained for further training of the DNN. Details on this training procedure and search for optimal values of $\beta$ are given in the Methods section of this article.

This architecture presents the advantage of enabling bias control via a unique hyperparameter. Moreover, both networks are naturally trained at the same time rather than in alternate, thus accelerating training approximately two-fold and enabling high-momentum gradient optimizers. To illustrate the architecture's output, we train the DNN on AM-sine waves and show a prediction example in S1 Fig in S1 Appendix, alongside the noisy input signal.

## 2.3 Post-training performance evaluation

We train the DNN on a random selection of decaying monochromatic sine waves (no modulation). The training, validation, and test samples are generated using random frequency, phase, coherence time, and noise levels. After training, we evaluate the DNN performance by comparing its prediction error to an LS-fit using the Python Scipy library. When performing the LS-fit, the input data is the noisy signals and the model function consists of the exact noiseless data-generating functions, given in Section 4.1. The LS-fit then produces predictions of the true latent parameters.

To this end, the LS-fit requires latent-parameters initial guesses to start the gradient descent. The initial guesses used here are the true latent parameters (i.e. true frequency, phase, and coherence time). After gradient descent, we use the LS-fit outputs to generate a prediction of the noiseless signals. This is done by inputting the LS-fit latent-parameters predictions in the data-generating process. The LS-fit and DNN performance are then compared in two ways: (i) the latent-parameters regression loss is the *MSE* from the true latent parameters for both the LS-fit and DNN ($MSE_{reg}$), and (ii) the denoising error is the *MSE* from the true noiseless signals for both the LS-fit and DNN ($MSE_{dec}$). Note that this comparison drastically favors the LS-fit, which then constitutes a good benchmark method. Indeed, in any practical applications the true value of the latent parameters are hidden from the user, and LS-fits are employed precisely to approximate them.

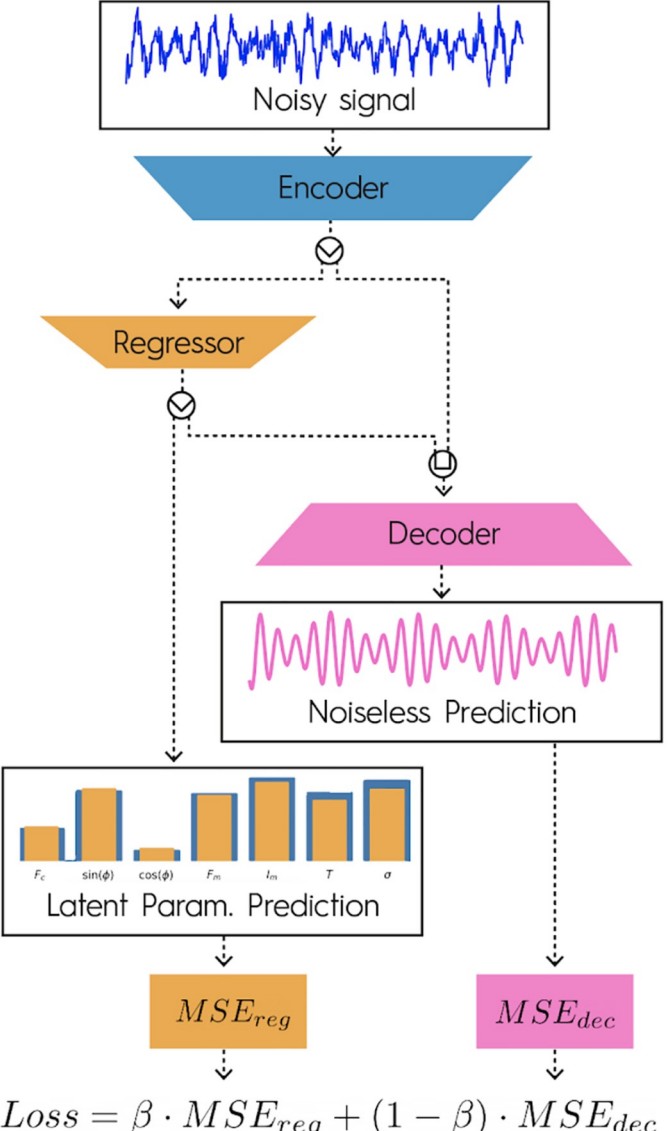

**Fig 2. Unified DNN architecture and loss description.** The *Encoder* produces a reduced representation of the input noisy signals. The *Encoder* output is passed to the *Regressor*, which outputs the latent parameters' prediction. The *Encoder* and *Regressor* outputs are passed to the *Decoder*, which produces a noiseless prediction of the inputs. The *Regressor* and *Decoder* outputs are used to compute the regression and denoising losses, $MSE_{reg}$ and $MSE_{dec}$, respectively. The loss used during backpropagation is a weighted sum of $MSE_{reg}$ and $MSE_{dec}$ using a bias parameter $\beta$.

A random selection of 1000 noisy signals from the test set is processed using this method. Fig 3 shows the relative $MSE_{reg}$ and $MSE_{dec}$ for both the DNN and LS-fit sorted by noise level (examples of signals with extremum noise levels, alongside LS-fit and DNN predictions are shown in S2 Fig in S1 Appendix). A similar evaluation is performed using AM-samples. In this experiment, the DNN is specifically trained on AM-samples. Examples of such samples are given in S2 Fig in S1 Appendix. S3 Fig in S1 Appendix shows the prediction errors of all samples, for both the DNN and LS-fit sorted by noise levels.

For both monochromatic and AM-signals, the DNN performs generally worse than the LS-fit for low-noise signals. Nonetheless, the DNN reaches LS-fit performance-level once the noise reaches the top half of the allowed range, while requiring no initial guesses. The latent-

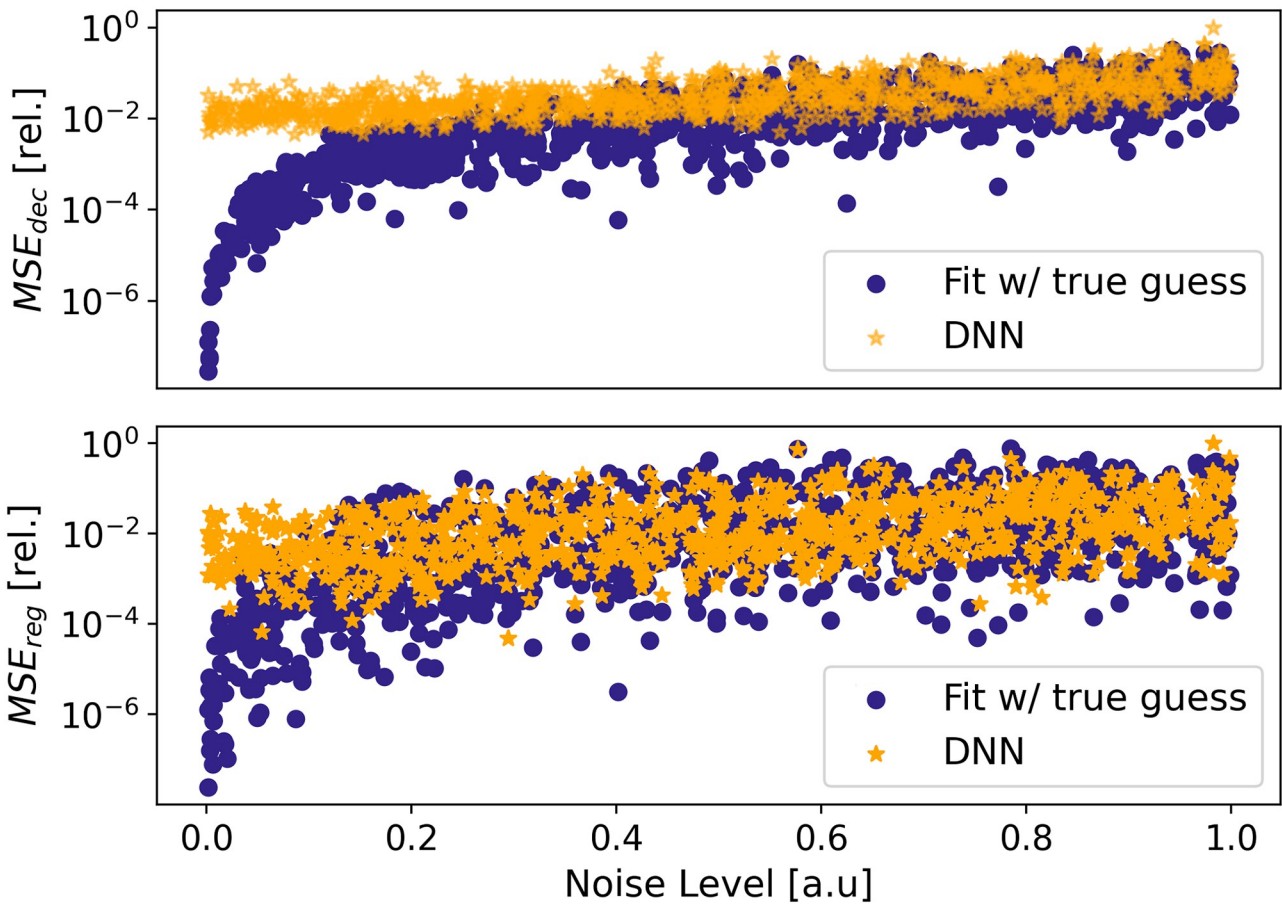

**Fig 3. Comparison of DNN post-training performance to LS-fits with true latent-parameters initial guesses for 1000 random monochromatic, decaying sine waves from the test set (unseen during training).** The denoising ($MSE_{dec}$, top) and latent-parameters relative regression losses ($MSE_{reg}$, bottom) are sorted by increasing noise levels. The DNN was trained on monochromatic sine waves samples. $MSE_{reg}$ is the MSE from the true latent parameters to the predicted latent parameters. For the DNN, $MSE_{dec}$ is the MSE from the true noiseless signal to the *Decoder* noiseless-signal prediction. For the LS-fits, $MSE_{dec}$ is computed similarly, but the noiseless-signal prediction is generated by inputting the predicted latent parameters in the noiseless data-generating process. The LS-fit with true initial guesses vastly outperforms the DNN for low-noise signals but both systems reach similar performance for high-noise.

parameters regression follows a similar trend. We note that, in general, DNN outputs are less sensitive to noise, and the performance is more consistent throughout both datasets.

Fitting oscillating time series using LS-fits is notoriously difficult because the *MSE* is in general a non-convex function of the latent parameters. As a result, poorly initialized LS-fits often remain trapped in local minima. Consequently, the quality of the LS-fit is highly dependent on the initial guesses in addition to the noise. For the same reason, random LS-fit initialization yields poor results with very high variability.

In any real-world setting, the user must perform additional preprocessing work or use prior information to find initial guesses leading to the global minima. When latent parameters can only exists within finite ranges (e.g. the signal phase must be constrained within $[0, 2\pi]$), a typical method involves performing a grid search over the space of initial guesses [16]. The LS-fit is repeatedly performed by using different values within the ranges. The LS-fit with the lowest final loss, is then assigned to be the best and its predictions are kept. Other methods involve performing a first estimation of the true latent parameters prior to performing the LS-fit (e.g. the signal's carrier frequency can be approximated by counting the numbers of local maxima

within the time windows or by finding the location of the maximum value of the signal's Fourier transform). Therefore, the complexity, runtime and usertime of LS-fits is highly dependent the amount of samples to analyze and the types of signals at hand.

In order to compare the speed performance of the DNN to real-world LS-fits (for which the initial guesses are unknown), we implemented an algorithm to perform a search over the initial space prior to the LS-fits. The samples are taken from the AM-sinewave validations sets. We first estimate the carrier frequency by finding the location of the power spectral density maximum. The signal phase is estimated by computing its gradient over the first half period. All other latent parameters are estimated by performing a grid search over their allowed ranged. The LS-fits function is then run for all initial guess combinations. Parameters yielding the lowest final loss are kept for the final LS-fit.

The average runtime for a single signal is on the order of 1 s (LS-fit gradient descent included). Thus, fitting an entire dataset (100′000 signals) yields a runtime on the order of one day. On the other hand, training the DNN took approximately one day and inference on the entire dataset takes only a few seconds.

As a result, in some cases, employing the DNN could prove faster and less complex. Indeed, initial guess exploration prior to LS-fits linearly depends on the number of samples to analyze and their length, whereas DNN training does not. Moreover, LS-fits performance and initial-guess exploration is highly dependent on the complexity and number of latent parameters of the data-generating model which is not the case for DNNs.

These results show that our architecture can be a good alternative to LS-fits for time-series analysis. First, the DNN reaches acceptable performance when benchmarked to standard LS-fits with true guesses, while needing no initial guesses (the DNN being initialized randomly prior to training). Moreover, the complex task of exploring the initial-guesses space for each sample during LS-fit is no longer needed when employing the DNN. This can automate and accelerate data processing.

## 2.4 DNN-assisted LS-fit

We now wish to apply our DNN in more realistic settings. In the previous experiments, LS-fits were only performed as a benchmark method, and the initial guesses were the true latent parameters. Here, we propose to employ the DNN as a preprocessing tool to assist LS-fit in the situation when the user possesses no prior information about the initial guesses and wishes to recover the signal's latent parameters. The sine wave samples from the previous experiment are fitted while using the DNN latent predictions as initial guesses. Results of this experiment are shown in Fig 4 alongside LS-fits with true initial guesses results.

Because the DNN predictions are always within the venicity of the true parameters, almost all DNN-assisted LS-fits converge to optimal solutions. In settings when the initial guesses are unknown or samples are numerous, the user can initially train the DNN on synthetic data and use it for DNN-assisted fits. As the latter performs optimally regardless of the noise level, this enables fast and accurate analysis of large datasets by automating the initial guesses exploration. Moreover, in addition to accelerate data processing, the use of DNNs to assist LS-fits can partially address the black-box character of DNNs. Indeed, this conjunction use of DNNs and LS-fits enables access to point-prediction uncertainties, typically in the form of covariance matrices, which DNNs typically lack [17–19].

## 2.5 Partial information regression and denoising

In the experiments presented above, the data-generating process was assumed to be fully known by the user. The DNN or DNN-assisted LS-fits were employed to recover the signal

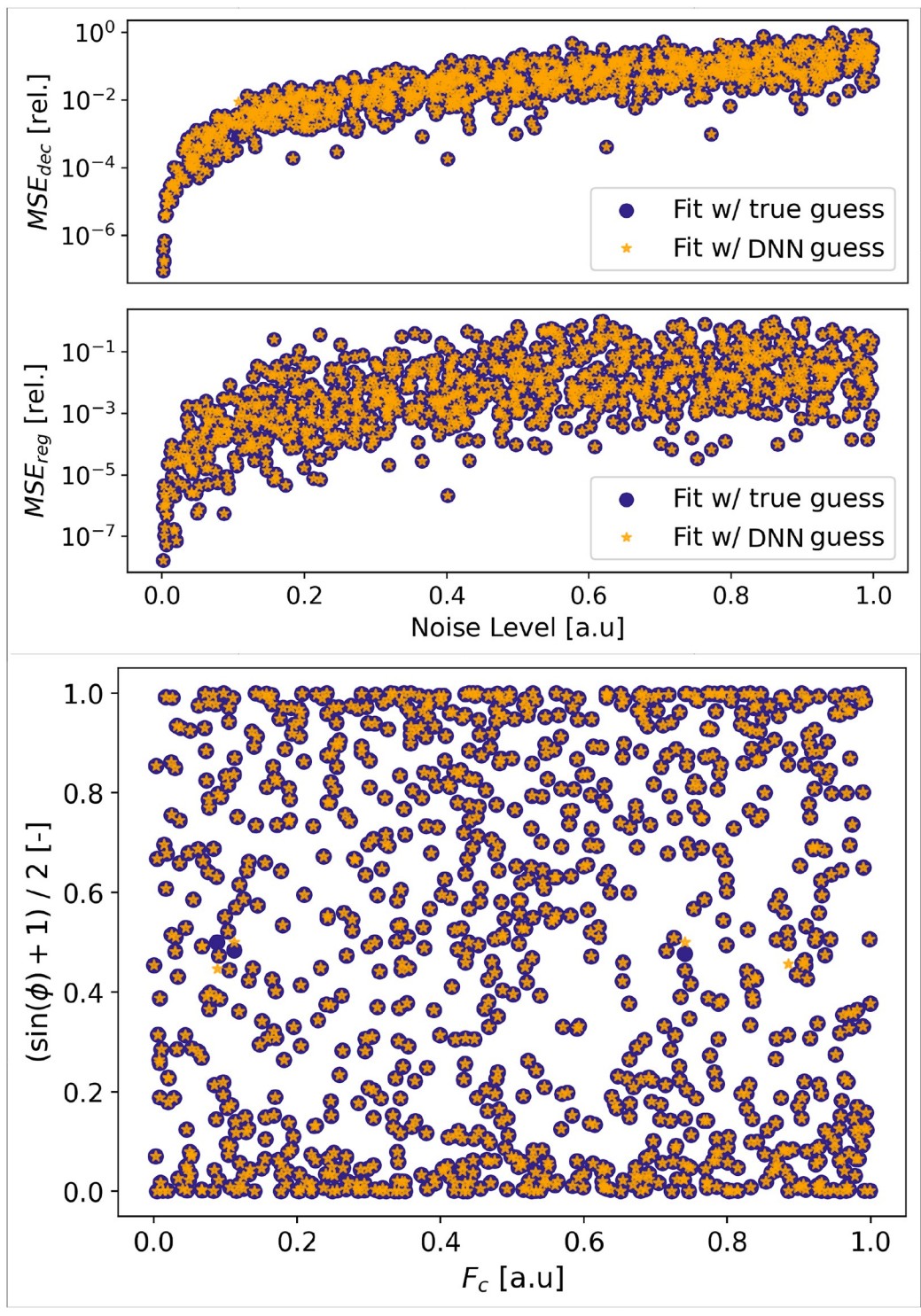

**Fig 4. DNN latent-parameters predictions used as initial guesses for DNN-assisted fits.** Comparison to LS-fit with true initial guesses. **Top**: The denoising ($MSE_{dec}$) and latent-parameters relative regression losses ($MSE_{reg}$) are sorted by increasing noise levels. See Fig 3 for $MSE_{reg}$ and $MSE_{dec}$ computation methods. **Bottom**: Phase and carrier frequency predictions for the DNN-assisted fits and LS-fits. Both methods converge to the same losses and predictions for over 99% of the samples. The DNN and data employed here are identical as in Fig 3.

latent parameters and denoise the signal. We now wish to explore the possibility of employing the DNN in a situation where the data-generating processes to be explored are multi-fold and guesses must be done. This is typically the case in "new-physics searches" experiments [13], during which hypothetical and undiscovered particles may cause signals deviating from the null-hypothesis (i.e. no new particles). As the hypothetical particles are numerous, they may have many potential effects on the signals. We take the situation in which a potential external source could modulate a carrier signal produced by the experiment, as it is sometimes the case for bosonic dark-matter [20].

Specifically, we study the case in which the end-user is aware of the existence of an oscillation in the signal provided by the experimental setup. The user ignores if the signal is monochromatic, amplitude or frequency modulated. Nonetheless, the user wishes to recover the frequency, phase, and coherence time of the expected oscillation.

In this situation, the typical approach is to test all allowed processes by varying the LS-fits objective functions and explore the space of initial guesses for each process. This approach presents a new set of challenges, as this exploration is time consuming and sometimes unrealistic, if the data is too large or if too many processes are to be tested. Moreover, in some situations, all guesses can be wrong.

We show that it is possible to perform the regression and denoising with partial prior information about the physical process producing the data. That is, the DNN is tasked to perform the regression only on the narrow set of latent parameters that exist across all models: frequency, phase, coherence time, and noise level. However, the DNN ignores any form of modulation. This is done by decreasing the number of neurons in the *Regressor*'s output layer. The DNN is then trained on signals from every explored model (monochromatic, AM and FM). We now refer to this DNN as the *partial DNN* (ignoring the existence of particular modulation type).

After training, we compare the performance of the partial DNN to a specialized DNN, trained specifically on AM signals, which performs a regression of all latent parameters. Fig 5 shows the $MSE_{reg}$ and $MSE_{dec}$ averaged over the AM-sine wave test set (100′000 samples) for both the AM-specialized DNN and the partial DNN. Because the partial DNN is trained on a wider variety of data, its prediction accuracy is, on average, lower than the specialized DNN. However, both DNNs performance remain close and the partial DNN is sufficiently accurate to perform the previously described experiments.

Using this method, the user's prior information is encoded into the *Regressor* architecture and training data. The *Regressor* then captures the expected latent parameters co-existing across the entire training set. The *Encoder* and *Decoder* remain unchanged and are still able to capture unknown latent parameters by reproducing noiseless signals. As a result, prior to the analysis, the user need not be fully aware of the data generating physical model but can instead train the DNN on a wider class of models (in this instance, the DNN was trained assuming the presence of any type of modulation). This feature is of particular importance in exploration of data obtained from new physics searches (such as dark-matter searches), in which the multitude of allowed physical models enables various forms of emerging signatures within the data. Thus, in addition to removing the need to iteratively explore models, this method enables weaker and partial prior information to be employed, while leaving space for signal exploration and unexpected discoveries.

## 3 Discussion

We have presented an efficient DNN that combines the denoising of times series and regression of their latent parameters. The DNN was trained and evaluated on synthetic monochromatic,

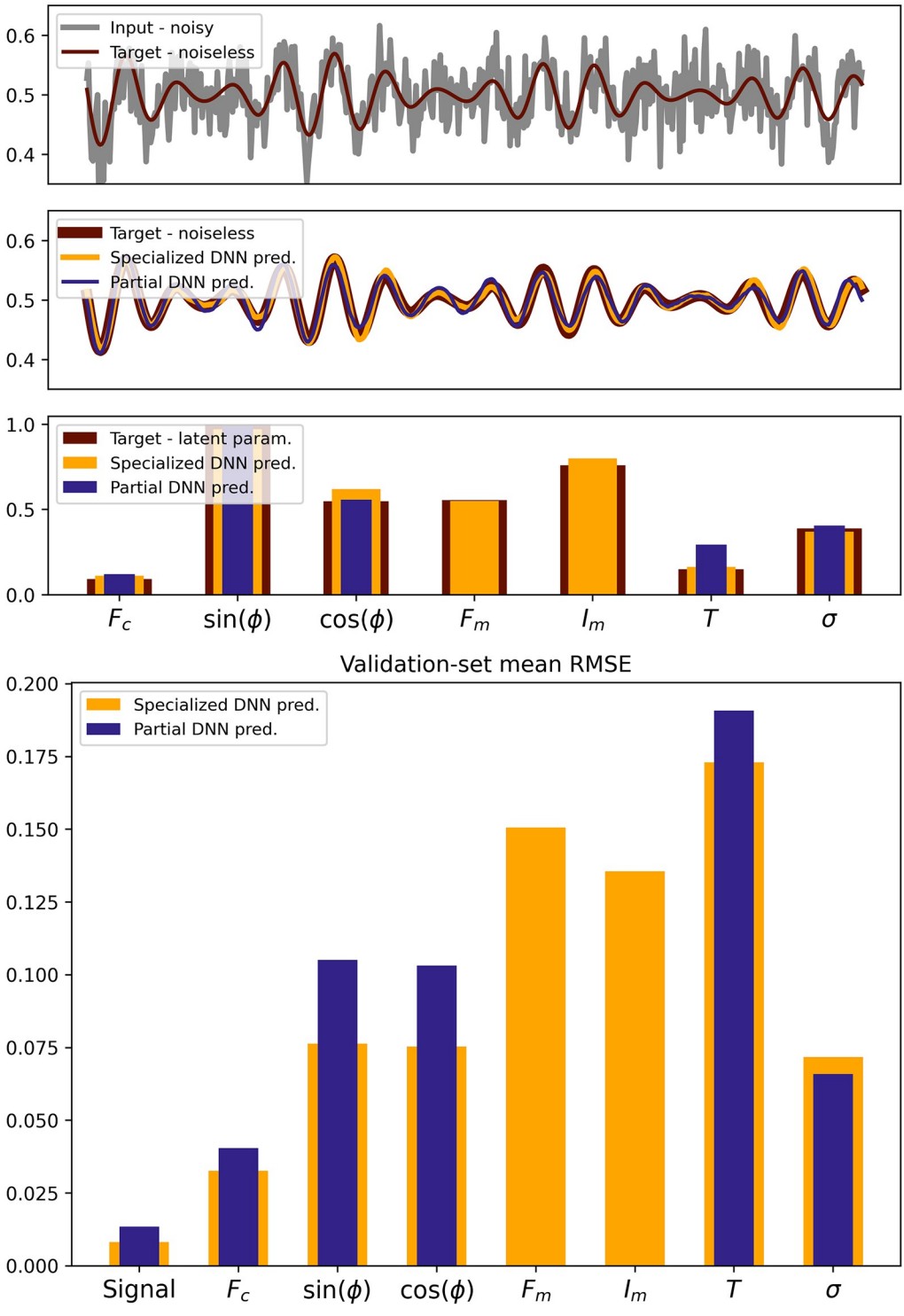

**Fig 5. Performance comparison of the specialized DNN (trained on AM-sine waves, tasked to denoise signals and recover *all* latent parameters of AM-sine waves) and of the partial DNN (trained on monochromatic, AM- and FM-sine waves, tasked to denoise signals and recover the carrier frequency, phase, coherence time and noise level only).** **Top**: Randomly selected example of noisy input AM-signal, alongside specialized- and partial-DNN denoised and latent predictions. **Bottom**: Individual latent parameters and signal denoising root mean squared error (RMSE), averaged over the whole AM-sinewave test set (100′000 samples) for both DNNs.

frequency- and amplitude-modulated decaying sine waves with Gaussian noise; some of the most prevalent forms of signals acquired in physics.

For high-noise signals, the DNN reaches same levels of precision as an LS-fit with true initial guesses, in spite of the DNN needing no guesses at all. In addition, the architecture requires no hyperparameter fine tuning to perform consistently. Moreover, because large volumes of synthetic training data can be generated, the DNN is quickly adaptable to a broad range of physical signals. This makes our architecture a good alternative to LS-fits for analysing large volumes of data, when fitting individual signals requires too much computation or user time.

The DNN architecture is flexible and can accommodate for various levels of user prior information. First, the DNN was used to assist LS-fits and predict initial guesses, unknown by the user. In this situation, DNN-assisted LS-fits consistently converge to the optimal solutions. Moreover, the system can be employed for signals of various lengths by adjusting the number of input and output neurons of the *Autoencoder*. For signals of arbitrary and varying length, or for online applications, a recurrent neural network architecture could also be implemented [21]. In both cases however, the end user must remain aware of the possibility of over- and under-sampling signals. Indeed, in the case of this work, the frequency resolution is limited by the number of input and output neurons of the *Autoencoder*. Moreover, the memory of a recurrent neural network is always finite, regardless of the signal length. The user must then adjust the allowed signal frequencies by restricting the latent parameters ranges employed during data generation.

Because training is done on arbitrarily large volumes of synthetic data, raw performance could be improved by increasing the number of trainable parameters such as adding more layers or neurons, without too much concern for overfitting. The architecture itself could be augmented by adding an upstream classifier DNN-module, which could identify the type of signals being analyzed. Classified signals could then be processed via specialized versions of our architecture, trained on the corresponding type of signals.

Time-domain oscillations generally appear as peaks or peak multiplets in frequency-domain spectra. Frequency, amplitude, and phase information is then localized to narrow regions of the spectral data. For that reason, we believe further improvements could be attained by making use of frequency-domain information. We suggest to use Fourier transforms or power spectra as DNN inputs, in addition to the raw time series.

Further performance improvements could also be obtained by using known constraints on the latent-parameters. In the case of this work, one could encode the Pythagorean identity to ensure the two-point phase prediction satisfies $\cos(\phi)^2 + \sin(\phi)^2 = 1$. Other constraints may include the expected allowed ranges of each latent variables. This is typically done by adding a cost to the loss function used during training. In our work, because the Pythagorean identity was violated on the same order of magnitude as the phase-prediction error, we did not implement such constraint but we nonetheless mention this as a technical possibility. Implementing constraints has been shown to improve performance and would further reduce the black-box character of DNNs by adding specific domain-knowledge to the model [22, 23].

The proposed DNN architecture can be used to detect and approximate hidden features in time series data. The *Regressor* outputs a prediction of prior known parameters, but real signals could still contain unknown latent variables. These hidden latent variables can be detected and approximated by our DNN, as it also incorporates an *Autoencoder*-like structure. As such, the bottleneck layer contains a feature representation of the time series, used by the *Decoder* to recreate the original signal. This bottleneck layer will be further investigated, in order to detect and specify hidden latent parameters.

We remain aware that in physics data analysis, a sole estimation of latent parameters often provides insufficient information. Standard analysis usually requires a quantitative estimation

of the prediction uncertainty, often represented as error bars or confidence intervals. In LS-fits, this uncertainty is naturally obtained by maximizing the fit likelihood under the assumption of Gaussian distributed latent variables [24]. Despite extensive efforts, DNNs still lack the capacity for reliable uncertainty evaluation [17–19]. While the DNN-assisted LS-fit method presented above partially solves this issue, more work needs to be done in this area to further generalize DNN usage in physics signal processing.

Nonetheless, we believe this architecture is readily applicable to existing physics experiments, in particular bosonic dark-matter searches [20, 25, 26], in which large quantities of data are to be analyzed with partial prior information.

## 4 Methods

### 4.1 Data generation procedure

The time series used throughout the article are generated by propagating the time, $t$, from 0 to 511 (with length $T = 512$) in 1 s increments, and using the following formula:

$$f_{Mono}(t) = \cos(2\pi F_c t + \phi) \cdot e^{-t/\tau},$$
$$f_{AM}(t) = \cos(2\pi F_c t + \phi) \cdot e^{-t/\tau} \cdot (1 + I_m \cos(2\pi F_m t)),$$
$$f_{FM}(t) = \cos\left(2\pi F_c t + \phi + \frac{0.01 I_m}{F_m}\cos(2\pi F_m t)\right) \cdot e^{-t/\tau},$$

where $F_c$ and $\phi$ are the sine wave carrier frequency and phase, respectively. $F_m$ and $I_m$ are the modulation frequency and amplitude. Noise is linearly added to the signals after being sampled from the Gaussian distribution with zero mean and standard deviation $\sigma$.

Before each sample generation, the latent parameters are randomly and uniformly sampled within the following ranges:

$$F_c \in [10/T, 1/8], \qquad \phi \in [0, 2\pi],$$
$$F_m \in [1/T, 1/16], \qquad I_m \in [0, 1],$$
$$\tau \in [0.2T, 8T], \qquad \sigma \in [0, 1].$$

Most DNN implementations generally require input and target data to be normalized such as to avoid exploding and vanishing gradients during training [27, 28]. All signals and latent parameters are normalized to lie within the 0-to-1 range prior to the application of the DNN. The phase $\phi$ is mapped to two separate parameters, $\phi :\longrightarrow \{\frac{\sin(\phi)+1}{2}; \frac{\cos(\phi)+1}{2}\}$, such as to account for phase periodicity during loss computation, while keeping both targets properly normalized. All other latent parameters are normalized using their respective range. Prior to adding the noise, we normalize the pure signals such that the resulting noisy signals remains within the [0, 1] range with mean Â 0.5. This normalization is performed identically for all signals.

### 4.2 DNN architecture and training procedure

The *Encoder* composed of 2[*Conv1D – Maxpool*] layers, followed by 2 *Dense* layers. The *Encoder* output layer has 64 neurons. The *Regressor* is composed of 2[*Conv1D – Maxpool*] followed by a *Conv1D* and 4 *Dense* layers. The output dimension of the *Regressor* is adjusted to the number of latent parameters that the *Regressor* is tasked to detect. The *Decoder* is composed of 1 *Dense*-3[*Conv1D-Maxpool-Upsampling*] layers, followed by a single *Conv1D* layer. The *Decoder* consists of a concatenation of the *Regressor* and *Encoder* ouputs.

All activation functions are rectified linear units, with the exception of the *Regressor* and *Decoder* outputs, which are linear and sigmoid function, respectively.

We unify the three architectures into a single DNN by passing the *Encoder* output to the *Regressor*, which predicts the signal's latent parameters. The *Decoder* input then consists of a concatenation of the *Regressor* and *Encoder* outputs.

During training, the *Regressor*'s target data consists of the latent parameters, and the *Decoder* target data are the noiseless signals. For both, the loss function is the mean squared error (*MSE*). The optimized architectures achieve sufficient performance, while keeping the number of trainable parameters under 1 million, such as to be able to perform training on a modern laptop GPU under 12 hours for a typical training session of 20 training sets of 100′000 samples, over 10 epochs. Due to the number and characteristics of the instances, asymptotic loss is reached within a small number of epochs. In general, increasing the number of training set instances was more beneficial than increasing the number of epochs.

To illustrate the effect of the bias parameter, we train the unified DNN on identical FM-sine waves datasets with varying values of $\beta$. For this experiment, training is performed using 12 training sets of 100′000 randomly generated samples for 10 epochs. Because the number of synthetic samples is large and the latent parameters are continuous random variables, overfitting (controlled by a validation set, unseen during training) was never an issue.

The performance of the trained DNN is evaluated using a test set of 100′000 randomly generated FM-samples, which were unseen during training. Fig 6 shows the test-sample losses for

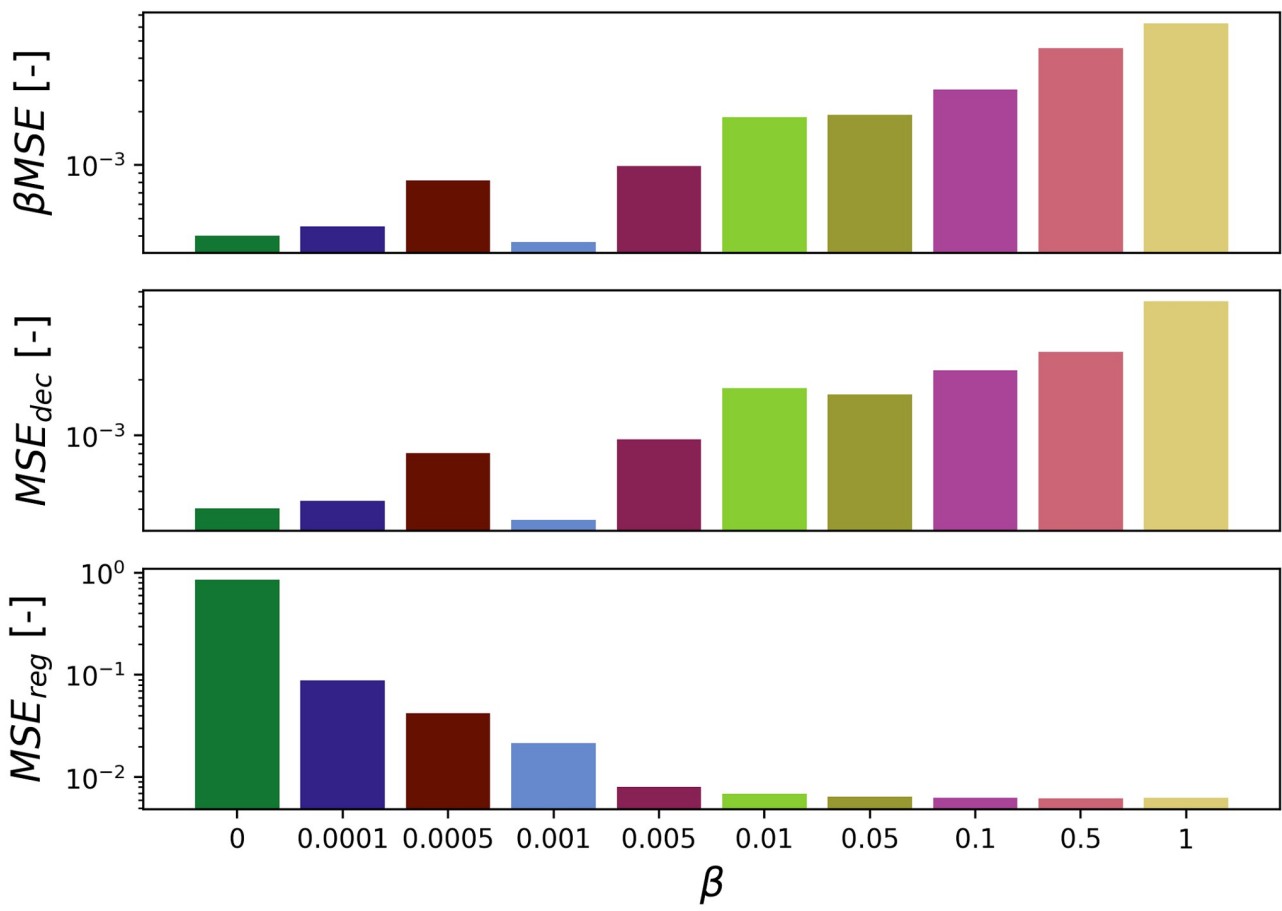

**Fig 6. FM-sine waves test-data prediction errors: Total weighted loss $\beta MSE$, denoising loss $MSE_{dec}$ and regression loss $MSE_{reg}$ for varying values of $\beta$.** Setting $\beta = 0$ or $\beta = 1$ fully biases training toward one of the two tasks, preventing the negatively-biased tasks to reach sufficient performance. Middle-range values enable both tasks to be learned simultaneously.

the denoising (top) and regression (bottom) tasks after training. Setting $\beta = 0$ fully biases training towards the denoising tasks, which achieves best performance, while the parameter regression yields the worst results; vice versa for $\beta = 1$. This behaviour is also observed in S4 Fig in S1 Appendix, which shows the validation losses during training. The training curves show that extremum values of $\beta$ prevents validation loss improvement of the negatively-biased task. Middle-range values enable both tasks to be learned simultaneously.

We find that the best values of $\beta$ are those for which the initial $\beta$-weighted regression and denoising losses are within the same order of magnitude. As a result, determining a good value for $\beta$ is a trivial task: A single forward pass is performed to obtain the initial values of $MSE_{reg}$ and $MSE_{dec}$. Regardless of the type of data (monochromatic-, AM- and FM-samples), DNNs trained with $\beta = 0.001$ achieve good overall performance (lowest weighted total loss) and little bias towards any of the tasks. This value of $\beta$ is employed throughout the entire article. For all that follows, training is always performed using 20 training sets of $100'000$ randomly generated samples for 10 epochs. This training is always enough to reach asymptotic loss, while exhibiting no noticeable overfitting. Training can be performed on decaying monochromatic-, AM-, FM-sine waves or a combination of all three processes.

## Supporting information

**S1 Appendix.**
(PDF)

## Acknowledgments

The authors wish to thank Jim Visschers for useful discussions and comments.

## Author Contributions

**Conceptualization:** Antoine Garcon, Julian Vexler, Dmitry Budker, Stefan Kramer.

**Data curation:** Antoine Garcon.

**Formal analysis:** Antoine Garcon, Stefan Kramer.

**Funding acquisition:** Antoine Garcon, Dmitry Budker, Stefan Kramer.

**Investigation:** Antoine Garcon, Julian Vexler, Dmitry Budker, Stefan Kramer.

**Methodology:** Antoine Garcon, Julian Vexler, Dmitry Budker, Stefan Kramer.

**Project administration:** Antoine Garcon, Dmitry Budker, Stefan Kramer.

**Supervision:** Dmitry Budker, Stefan Kramer.

**Validation:** Julian Vexler.

**Writing – original draft:** Antoine Garcon, Julian Vexler.

**Writing – review & editing:** Antoine Garcon, Julian Vexler, Dmitry Budker, Stefan Kramer.

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
