## [Decision Letter · Decision Letter 0]

14 Dec 2021

PONE-D-21-24201Deep neural networks to recover unknown physical parameters from oscillating time seriesPLOS ONE

Dear Dr. Garcon,

Thank you for submitting your manuscript to PLOS ONE. After careful consideration, we feel that it has merit but does not fully meet PLOS ONE’s publication criteria as it currently stands. Therefore, we invite you to submit a revised version of the manuscript that addresses the points raised during the review process.

 Please carefully address all the comments raised by both the reviewers

We look forward to receiving your revised manuscript.

Kind regards,

Sheetal Kalyani

Academic Editor

PLOS ONE

Journal Requirements:

“This work was supported in part by the Cluster of Excellence PRISMA+ funded by the German Research Foundation (DFG) within the German Excellence Strategy (Project ID 39083149), by the European Research Council (ERC) under the European Union Horizon 2020 research and innovation program (project Dark-OST, grant agreement No 695405), and by the DFG Reinhart Koselleck project. A.G. acknowledges funding from the Emergent AI Center funded by the Carl-Zeiss-Stiftung”

“This work was supported in part by the Cluster of Excellence PRISMA+

funded by the German Research Foundation (DFG) within the German Excellence

Strategy (Project ID 39083149), by the European Research Council (ERC) under the

European Union Horizon 2020 research and innovation program (project Dark-OST,

grant agreement No 695405), and by the DFG Reinhart Koselleck project. A.G.

acknowledges funding from the Emergent AI Center funded by the Carl-Zeiss-Stiftung.

4. Please include a caption for figure 6.

Reviewers' comments:

Reviewer's Responses to Questions

**Comments to the Author**

1. Is the manuscript technically sound, and do the data support the conclusions?

Reviewer #1: Yes

Reviewer #2: Yes

2. Has the statistical analysis been performed appropriately and rigorously? 

Reviewer #1: I Don't Know

Reviewer #2: I Don't Know

3. Have the authors made all data underlying the findings in their manuscript fully available?

Reviewer #1: Yes

Reviewer #2: Yes

4. Is the manuscript presented in an intelligible fashion and written in standard English?

Reviewer #1: Yes

Reviewer #2: Yes

5. Review Comments to the Author

Reviewer #1: - In Fig 2, two latent parameters are used for learn about \\phi. How the trigonometric relationship between these components is ensured? It is, how is it ensured in the model that sin^2 \\phi + cos^2 \\phi = 1?

- How is MSE_reg computed? A detailed description in text is required.

- \\beta parameter is assumed to help in bias control of the model (Pg 5, last paragraph). Please give detailed description of how this is achieved by tuning the \\beta parameter.

- For LS-fit, please provide the exact model used.

- How good is LS-fit performance with random initialization followed by gradient descent training (similar to DNNs)?

- It is not clear why deep learning is better than LSfit method for the problems discussed in the manuscript. One of the points that is consistently repeated is that LSfit is required to know an initial point. However the same goes with DNNs as well. Please clarify how a randomly initialized DNN is better than LSfit in this case?

- The proposed model is designed to work for only specific length sequences (512 in the presented experiments). However, with the given sampling specification, this may under-sample the signal which might result in missing the signal. Please provide a description on the limitations of sampling/network input and how this could be handled.

- One of the ways to extent the architecture to handle arbitrary length sequences is to use recurrent neural network based encoder-decoder framework. It may be useful for the reader if a description pointing towards this direction is included.

Reviewer #2: The authors propose a deep learning model which combines an autoencoder and a regressor to estimate the latent parameters from an oscillating time series. The proposed idea where the output of the DNN is used as an initial estimate for Ls-fit is interesting. The authors' extension to estimating the parameters without knowing the exact data generation process was also impressive. Please find detailed comments listed below.

1. My major concern is the lack of literature review regarding how the latent parameter recovery is performed using traditional signal processing tools; in particular, the state-of-the-art method for this specific task.

2. The authors provide all their comparisons with LS-fit where the true parameters are known. As pointed out, this is not a fair comparison as the true parameters are generally unknown. The authors are encouraged to provide comparisons with methods that do not need to know the true parameters. If such a method involves a search over the parameter space followed by LS-fit, it is recommended that the authors compare complexity (in addition to performance).

3. The authors state in their abstract (and introduction) that physicists require quantitative information regarding their systems which is not provided by neural networks (which acts as a black box model). However, they do not address this issue in their work. The proposed method is also obtained from abstract data and does not provide interpretable insights regarding the learnt model.

4. A few details in results seem to be missing. For example, Figures 5 (Top) shows the noisy input signal along with the predicted parameters. Indicate if this is the best or the average behavior out of the 100000 samples.

6. PLOS authors have the option to publish the peer review history of their article (what does this mean?). If published, this will include your full peer review and any attached files.

Reviewer #1: No

Reviewer #2: No

---

## [Author Response · Author response to Decision Letter 0]

29 Mar 2022

Reviewer 1, comment 1/8:

In Fig 2, two latent parameters are used for learn about ϕ. How the trigonometric relationship

between these components is ensured? It is, how is it ensured in the model that sin2ϕ+cos2ϕ = 1 ?

Answer:

The Pythagorean identity was not used as a constraint during this work. After investigation, we

measured that this identity was violated on the same order of magnitude as the error on the

phase prediction itself. Nonetheless, we thank the reviewer for this suggestion and now discuss the

potential use of latent parameters constraints in the Conclusion section of this article, in addition

to discussing the reviewer’s idea to use recurrent neural networks for arbitrarily long signals (see

reviewer 1, comment 8/8). The said paragraph now reads:

"Further performance improvements could also be obtained by using known constraints on the

latent-parameters. In the case of this work, one could encode the Pythagorean identity to ensure

the two-point phase prediction satisfies cos(ϕ)2 + sin(ϕ)2 = 1. Other constraints may include the

expected allowed ranges of each latent variables. This is typically done by adding a cost to the loss

function used during training. In our work, because the Pythagorean identity was violated on the

same order of magnitude as the phase-prediction error, we did not implement such constraint but

we nonetheless mention this as a technical possibility. Implementing constraints has been shown to

improve performance and would further reduce the black-box character of DNNs by adding specific

domain-knowledge to the model [22,23]."

------

Reviewer 1, comment 2/8:

How is MSEreg computed? A detailed description in text is required.

Answer:

The computation technique was indeed not described. A short sentence has been added above Eq. 1

to clarify :

"The latent-parameter regression loss, MSEreg, and signal-denoising losses, MSEdec, are computed

simultaneously. MSEreg is the mean squared error between the DNN predictions and true latent

parameters, while MSEdec is the mean squared error between the denoised signal prediction and

the noiseless signal."

------

Reviewer 1, comment 3/8:

β parameter is assumed to help in bias control of the model (Pg 5, last paragraph). Please give

detailed description of how this is achieved by tuning the β parameter.

Answer:

We now include a short description of the tuning of β in the main text:

"The value of β is determined by performing a grid search over the [0, 1] range: We train a new

instance of the DNN for varying values of β. The best value (yielding the lowest overall loss and

low bias towards any of the tasks, determined on a validation set) is maintained for further training

of the DNN. Details on this training procedure and search for optimal values of β are given in the

Methods section of this article."

A more detailed discussion of the effect of the bias parameter is included in the Methods section,

supported by Fig.S4 & Fig.6, which illustrate the effect of this hyper-parameter during and after

training.

------

Reviewer 1, comment 4/8:

For LS-fit, please provide the exact model used.

Answer:

The model used were the exact generating functions for the AM, FM and monochromatic sinewave,

given in the Materials and Method. This was not clearly stated in section 2.3 which now reads:

"When performing the LS-fit, the input data is the noisy signals and the model function consists of

the exact noiseless data-generating functions, given in Section 4.1."

------

Reviewer 1, comment 5/8:

How good is LS-fit performance with random initialization followed by gradient descent training

(similar to DNNs)?

Answer:

Since both reviewers 1&2 had similar concerns about LS-fits initialization and performance, we

addressed them together by entirely rewriting section 2.3. We now indicate within section 2.3:

1- The poorness of the outcome for LS-fit with random initialization (reviewer 1 comment 5)

2- A description of state-of-the-art signal processing tools to recover latent parameters (reviewer 2

comments 1 & 2)

3- We explain that the DNN-assisted fit can be an interesting technique to automate searching the

space of initial guesses prior to LS-fits (reviewer 2 comment 2)

------

Reviewer 1, comment 6/8:

It is not clear why deep learning is better than LSfit method for the problems discussed in the

manuscript. One of the points that is consistently repeated is that LSfit is required to know an

initial point. However the same goes with DNNs as well. Please clarify how a randomly initialized

DNN is better than LSfit in this case?

Answer:

We feel that our presentation needed improvement, in particular of the DNN and LS-fit initializations.

Please note the following:

a) On the DNN and LS-fits initialization: In this work, the DNN is always initialized at random

prior to training (by assigning random weights to each neuron). LS-fits, however, require "good" initial

points, typically referred to as "initial guesses", which must be close to the true latent parameters.

b) On the potential advantage of using the DNN: The advantage of the neural network approach is

then to automate the exploration of the initial space performed during LS-fits (which is replaced by

training the DNN on a general dataset).

These statements are made clearer in section 2.3:

"These results show that our architecture can be a good alternative to LS-fits for time-series

analysis. First, the DNN reaches acceptable performance when benchmarked to standard LS-fits

with true guesses, while needing no initial guesses (the DNN being initialized randomly prior to

training). Moreover, the complex task of exploring the initial-guesses space for each sample during LSfit

is no longer needed when employing the DNN. This can automate and accelerate data processing."

In addition, we modified the legend of Fig. 4 such as to make clearer the fact that the DNN is used

to assist LS-fits initial guesses exploration. We hope that this will avoid confusion about the DNN

initialization and advantages. It was indeed misleading before.

------

Reviewer 1, comment 7/8:

The proposed model is designed to work for only specific length sequences (512 in the presented

experiments). However, with the given sampling specification, this may under-sample the signal which

might result in missing the signal. Please provide a description on the limitations of sampling/network

input and how this could be handled.

Answer:

We ensure the signals is never over/undersampled by adjusting the range of allowed signal frequencies

during data generation. We modified Section 2.1 to now state this clearly:

"Before each sample generation, the latent parameters are randomly and uniformly sampled within

their respective allowed range (see Methods). The range of Fc ensures the carrier frequency

remains well within the Fourier and Nqyist limits such that no over- or under-sampling occurs. The

modulation amplitude range ensures the majority of the signal’s power remains in its first sidebands

and carrier."

Moreover, the experiment can be run with any signal length by adjusting the number of input

neurons of the encoder and output neurons of the decoder. In addition, we added a short statement

concerning this topic in the conclusion of the article (see also reviewer’s suggestion below).

------

Reviewer 1, comment 8/8:

One of the ways to extent the architecture to handle arbitrary length sequences is to use recurrent

neural network based encoder-decoder framework. It may be useful for the reader if a description

pointing towards this direction is included.

Answer:

We thank reviewer 1 for this suggestion, which we added in the conclusion section of the article:

"[...]. Moreover, the system can be employed for signals of various lengths by adjusting the number

of input and output neurons of the Autoencoder. For signals of arbitrary and varying length, or

for online applications, a recurrent neural network architecture could also be implemented [21]. In

both cases however, the end user must remain aware of the possibility of over- and under-sampling

signals. Indeed, in the case of this work, the frequency resolution is limited by the number of input

and output neurons of the Autoencoder. Moreover, the memory of a recurrent neural network is

always practically finite, regardless of the signal length. The user must then adjust the allowed

signal frequencies by restricting the latent parameters’ ranges employed during data generation."

------

Reviewer 2, general comment:

The authors propose a deep learning model which combines an autoencoder and a regressor to

estimate the latent parameters from an oscillating time series. The proposed idea where the output

of the DNN is used as an initial estimate for Ls-fit is interesting. The authors’ extension to estimating

the parameters without knowing the exact data generation process was also impressive. Please find

detailed comments listed below.

Answer:

We thank reviewer 2 for their kind comment.

------

Reviewer 2, comment 1/4:

My major concern is the lack of literature review regarding how the latent parameter recovery is

performed using traditional signal processing tools; in particular, the state-of-the-art method for

this specific task.

Answer:

Typically this task requires expertise on the side of the trained engineer or physicist. A search over

the initial guesses is manually performed by repeating LS-fits with different initial guesses. We now

describe the typical approach to time-series regression in section 2.3 which has been entirely rewritten:

"Fitting oscillating time series using LS-fits is notoriously difficult because the MSE is in general a

non-convex function of the latent parameters. As a result, poorly initialized LS-fits often remain

trapped in local minima. Consequently, the quality of the LS-fit is highly dependent on the initial

guesses in addition to the noise. For the same reason, random LS-fit initialization yields poor results

with very high variability [6].

In any real-world setting, the user must perform additional preprocessing work or use prior information

to find initial guesses leading to the global minima. When latent parameters can only exist within

finite ranges (e.g. the signal phase must be constrained within [0, 2π[), a typical method involves

performing a grid search over the space of initial guesses [16]. The LS-fit is repeatedly performed

by using different values within the ranges. The LS-fit with the lowest final loss, is then assigned to

be the best and its predictions are kept. Other methods involve performing a first estimation of

the true latent parameters prior to performing the LS-fit (e.g. the signal’s carrier frequency can

be approximated by counting the numbers of local maxima within the time windows or by finding

the location of the maximum value of the signal’s Fourier transform). Therefore, the complexity,

runtime and usertime of LS-fits is highly dependent the amount of samples to analyze and the types

of signals at hand."

------

Reviewer 2, comment 2/4:

The authors provide all their comparisons with LS-fit where the true parameters are known. As

pointed out, this is not a fair comparison as the true parameters are generally unknown. The authors

are encouraged to provide comparisons with methods that do not need to know the true parameters.

If such a method involves a search over the parameter space followed by LS-fit, it is recommended

that the authors compare complexity (in addition to performance).

Answer:

The reviewer is correct in stating that most methods involve a search over the initial guesses space,

followed by an LS-fit (see also reviewer’s previous suggestion above). The complexity of such

searches is entirely dependent on the situation at hand. Indeed, the complexity of the task would

depend on the types of signals to analyze, possible access to initial guesses and most importantly

on the number of samples to analyze.

To address this comment, we implemented an LS-fit algorithm to perform a search over the initial

space and compared its runtime to that of the DNN. The algorithm employs a combination of

expert-knowledge to estimate some latent parameters, and performs a grid search for parameters

that are more difficult to access. The average runtime for a single signal is on the order of 1 s

(LS-fit gradient descent included). Thus, fitting an entire dataset (100000 signals) yields a runtime

on the order of one day. On the other hand, training the DNN took approximately one day and

inference on the entire dataset takes only a few seconds. Thus, for a large number of samples or for

complex data-generating models, employing the DNN as a regression tool could prove faster than

repeating LS-fits.

For this reason we now state in section 2.3:

"[...]. In order to compare the runtime performance of the DNN to real-world LS-fits (for which the

initial guesses are unknown), we implemented an algorithm to perform a search over the initial space

prior to the LS-fits. The samples are taken from the AM-sinewave validation sets. We first estimate

the carrier frequency by finding the location of the power spectral density maximum. The signal

phase is estimated by computing its gradient over the first half period. All other latent parameters

are estimated by performing a grid search over their allowed ranged. The LS-fits function is then

run for all initial guess combinations. Parameters yielding the lowest final loss are kept for the final

LS-fit.

The average runtime for a single signal is on the order of 1 s (LS-fit gradient descent included).

Thus, fitting an entire dataset (100′000 signals) yields a runtime on the order of one day. On the

other hand, training the DNN took approximately one day and inference on the entire dataset takes

only a few seconds.

As a result in some cases, employing the DNN could prove faster and less complex. Indeed, initial

guess exploration prior to LS-fits linearly depends on the number of samples to analyze and their

length, whereas training DNN does not. Moreover, LS-fits performance and initial-guess exploration

is highly dependent on the complexity and number of latent parameters of the data-generating

model, which is not the case for DNNs.

These results show that our architecture can be a good alternative to LS-fits for time-series analysis.

First, the DNN reaches acceptable performance when benchmarked to standard LS-fits with true

guesses, while needing no initial guesses (the DNN being initialized randomly prior to training).

Moreover, the complex task of explorating the initial-guesses space for each individual sample being

replaced by training the DNN, can automate and accelerate data processing."

------

Reviewer 2, comment 3/4:

The authors state in their abstract (and introduction) that physicists require quantitative information

regarding their systems which is not provided by neural networks (which acts as a black box model).

However, they do not address this issue in their work. The proposed method is also obtained from

abstract data and does not provide interpretable insights regarding the learnt model.

Answer:

We feel that our method, while not fully resolving this issue, is still a step toward the right direction.

Indeed, we first employ a neural network to fully recover latent parameters of time-series, in addition

to denoise the actual signals. While being far from a fully explainable AI or a Physics learner, this

method still provides additional quantitative information (latent parameters estimation) than simply

outputting a denoised DNN prediction.

Moreover, the use of DNN in time-series regression is typically frowned upon in physics due to the

complication of accessing point-estimation uncertainty. Here, we propose to use the DNN to inform

LS-fits initial guesses, thus enabling access to a covariance matrix and uncertainty estimation.

We do feel that this was not clearly stated in section 2.4, which was modified by adding the following:

"Because the DNN predictions are always within the venicity of the true parameters, almost all

DNN-assisted LS-fits converge to optimal solutions. In settings when the initial guesses are unknown

or samples are numerous, the user can initially train the DNN on synthetic data and use it for

DNN-assisted fits. As the latter performs optimally regardless of the noise level, this enables fast

and accurate analysis of large datasets by automating the initial guesses exploration. Moreover, in

addition to accelerated data processing, the use of DNNs to assist LS-fits can partially mitigate the

black-box character of DNNs. Indeed, this conjunction use of DNNs and LS-fits enables access to

point-prediction uncertainties, typically in the form of covariance matrices, which DNNs typically

lack [17-19]."

------

Reviewer 2, comment 4/4:

A few details in results seem to be missing. For example, Figures 5 (Top) shows the noisy input

signal along with the predicted parameters. Indicate if this is the best or the average behavior out

of the 100000 samples.

Answer:

The signal was selected at random among the 100000 samples. Performance for this particlar sample

was averaged. This is now clearly stated in the figure caption.

---

## [Decision Letter · Decision Letter 1]

1 May 2022

Deep neural networks to recover unknown physical parameters from oscillating time series

PONE-D-21-24201R1

Dear Dr. Garcon,

We’re pleased to inform you that your manuscript has been judged scientifically suitable for publication and will be formally accepted for publication once it meets all outstanding technical requirements.

Kind regards,

Sheetal Kalyani

Academic Editor

PLOS ONE

Additional Editor Comments (optional):

Reviewers' comments:

Reviewer's Responses to Questions

**Comments to the Author**

1. If the authors have adequately addressed your comments raised in a previous round of review and you feel that this manuscript is now acceptable for publication, you may indicate that here to bypass the “Comments to the Author” section, enter your conflict of interest statement in the “Confidential to Editor” section, and submit your "Accept" recommendation.

Reviewer #1: All comments have been addressed

Reviewer #2: All comments have been addressed

2. Is the manuscript technically sound, and do the data support the conclusions?

Reviewer #1: Partly

Reviewer #2: Yes

3. Has the statistical analysis been performed appropriately and rigorously? 

Reviewer #1: Yes

Reviewer #2: I Don't Know

4. Have the authors made all data underlying the findings in their manuscript fully available?

Reviewer #1: Yes

Reviewer #2: Yes

5. Is the manuscript presented in an intelligible fashion and written in standard English?

Reviewer #1: Yes

Reviewer #2: Yes

6. Review Comments to the Author

Reviewer #1: The authors have addressed most of the comments raised in the previous round. However, it will be beneficial to discuss the below point in details: It is mentioned in the response that the constraint of sin \\phi, cos \\phi is violated, but with less magnitude, in the current model. It will be useful go give the magnitude of the error to give readers an idea bout this violation. Further, even after adding a soft constraint, this violation can still happen.

Reviewer #2: (No Response)

7. PLOS authors have the option to publish the peer review history of their article (what does this mean?). If published, this will include your full peer review and any attached files.

Reviewer #1: **Yes: **Vishnu Raj

Reviewer #2: No

---

## [Editor Report · Acceptance letter]

5 May 2022

PONE-D-21-24201R1 

Deep neural networks to recover unknown physical parameters from oscillating time series 

Dear Dr. Garcon:

I'm pleased to inform you that your manuscript has been deemed suitable for publication in PLOS ONE. Congratulations! Your manuscript is now with our production department. 

Kind regards, 

on behalf of

Dr. Sheetal Kalyani 

Academic Editor

PLOS ONE